# Integrated Analysis of Ginsenoside Content and Biomarker Changes in Processed Ginseng: Implications for Anti-Cancer Mechanisms

**DOI:** 10.3390/foods13162497

**Published:** 2024-08-08

**Authors:** Biyu Guo, Yingli Liang, Biru Fu, Jiayi Luo, Xingchen Zhou, Ruifeng Ji, Xin He

**Affiliations:** 1School of Traditional Chinese Materia Medica, Guangdong Pharmaceutical University, Guangzhou 510006, China; guobiyu233@163.com (B.G.); liangyingli1218@163.com (Y.L.); 17857314822@163.com (B.F.); logaga2@163.com (J.L.); jiruifeng@gdpu.edu.cn (R.J.); 2Jingji (Guangzhou) Biotechnology Co., Ltd., Guangzhou 510006, China; xz1469@nyu.edu.com

**Keywords:** cultivated ginseng, forest ginseng, processing, rare ginsenosides, UPLC-MS, chemometric

## Abstract

Black ginseng is the processed product of ginseng, and it has been found that the content and types of rare ginsenosides increased after processing. However, there is limited research on the ginsenoside differences between cultivated and forest ginseng before and after processing and among various plant parts. This study investigated the effects of processing on ginsenosides in different parts of cultivated and forest ginseng. After processing, the contents of Re, Rg1, S-Rg3, Rg5, R-Rh1, Rk1, Rk3, and F4 were significantly increased or decreased, the growth age of forest ginseng was not proportional to the content of ginsenosides, and the differences in ginsenoside content in ginseng from different cultivation methods were relatively small. Chemometric analysis identified processing biomarkers showing varying percentage changes in different parts. Network pharmacology predicted the EGFR/PI3K/Akt/mTOR pathway as a potential key pathway for the anti-cancer effect of black ginseng.

## 1. Introduction

Ginseng (*Panax ginseng* C.A. Meyer), affiliating with the genus Panax L. (Araliaceae), is known for its tonic properties, including replenishing vital energy, tonifying the spleen and lungs, nourishing body fluids, and promoting mental clarity [1]. Due to variations in growth environments and cultivation practices, the Chinese Pharmacopoeia includes two varieties of ginseng: cultivated ginseng and forest ginseng [2]. The morphological characteristics and chemical composition of various parts of ginseng, including the leaf, stem, rhizome, root, etc., can exhibit variations under different habitats, growth patterns, and growth ages [3,4,5,6]. These differences in chemical composition may contribute to the differential pharmacological effects exhibited by different parts of ginseng [7,8].

According to the processing methods, ginseng can be categorized into fresh ginseng, white ginseng, red ginseng, and black ginseng [9]. Currently, the market predominantly provides white ginseng and red ginseng, leading to a greater focus on research related to these types. Due to these market forces, there is a relatively limited amount of research on black ginseng [10]. Black ginseng is a processed product obtained by multiple rounds of steaming and drying of fresh ginseng [11]. Previous studies have suggested that black ginseng exhibits elevated quantities of rare ginsenosides after the processing, which generate its enhanced activities in anti-tumor [12], hypoglycemic [13], antioxidant [14], and anti-inflammatory [15] effects.

Prototype ginsenosides serve as the main pharmacologically active components and also function as the main indicators for evaluating the quality of ginseng [16]. According to the different aglycones, prototype ginsenosides can be divided into protopanaxadiol (PPD), protopanaxatriol (PPT), and oleanolic acid (OA) types [1]. Prototype ginsenoside is triterpene saponins composed of aglycones and glycosides, mainly present in fresh and white ginseng. However, after prolonged steaming and drying, the aglycone steroid backbone or glycoside side chain of the ginsenosides is changed to form rare ginsenosides [17]. This leads to differences in the physiological activities of various processed ginseng. Under high temperature and pressure, prototype ginsenosides (Rb1, Rc, Re, Rg1, etc.) can be transformed into rare ginsenosides (Rg3, Rg5, Rk1, F4, etc.) through depropanoylation, decarboxylation, deglycosylation, and dehydration reactions at the C-20 of the dammarane skeleton. The structural transformation is illustrated in Figure 1. Previous studies have found that Rg3, Rg5, and Rk1, with the highest content in black ginseng, played a major role in exerting anti-tumor and antioxidant effects [18]. The contents of all ginsenoside components were found to be significantly different among landraces and varieties, particularly for Rg1, Rb1, Rd, and 20(S)-ginsenoside Rg3 [19].

Studies have shown that the content of PPD-type ginsenosides in forest ginseng is higher than that in cultivated ginseng [20]. Additionally, the age of the ginseng plant also affects the content of ginsenoside in forest ginseng [21]. However, there is limited research on the differences in composition between cultivated ginseng and forest ginseng [22], both before and after processing, as well as the differences among various parts of the plant. In addition, the changes in the main active ingredients contained therein need to be analyzed in depth. Hence, the analysis of these related regularities will therefore offer a basis for improving subsequent studies on ginseng and black ginseng.

The emerging field of network pharmacology facilitates the establishment of associations among compounds, targets, and diseases, thereby offering novel strategies and methodologies for the revitalization of traditional Chinese medicine (TCM) research [23]. By employing network pharmacology, it becomes possible to anticipate the mechanisms through which the compounds demonstrate their effects on the organism, acquire their respective targets, and establish connections within the network.

In recent years, ultra-performance liquid chromatography coupled with triple-quadrupole tandem mass spectrometry (UPLC-MS/MS) and network pharmacology have emerged as prominent techniques for the analysis of bioactive compounds, molecular targets, and pharmacodynamics of TCM [24]. In this study, a sum of eleven prototype ginsenosides (such as ginsenoside Rb1) and nine rare ginsenosides (such as ginsenoside S-Rg2) were screened as the measured indicator components. The content of the indicator components in different parts of cultivated ginseng and forest ginseng before and after processing (including the leaf, stem, rhizome, main root, and lateral root) was determined by UPLC-MS/MS. Chemometrics methods were employed to analyze the biomarkers after processing and their variation in different ginseng samples. Finally, network pharmacology analysis was conducted to explore the anti-tumor effects of different biomarkers. The overall workflow is illustrated in Figure 2.

## 2. Materials and Methods

### 2.1. Plant Materials

A total of 36 samples of cultivated ginseng and forest ginseng were collected, with each sample comprised of five distinct parts: leaf, stem, rhizome, main root, and lateral root, as depicted in Figure 3. All samples were collected from Benxi City, Liaoning Province, China, in May 2023. Damaged or diseased samples were then removed, and 12 samples of each variety of *P. ginseng* were randomly divided into two groups. The samples were washed to remove dirt and excess moisture and then weighed after drying with filter paper. The white-ginseng group (WG) was dried at 60 °C, and based on previous research by the research team, the black-ginseng group (BG) was steamed at 100 °C for 2 h in a high-pressure sterilizer (Shanghai Shen’an Medical Appliance Factory, China), followed by drying at 60 °C for 12 h. The process for the BG was repeated 9 times, with the final drying continuing until the moisture content was less than 12%.

### 2.2. Chemicals and Reagents

Twenty commercially available authentic standards (Appendix A), with purity greater than 98%, were purchased from Chengdu Must Bio-Technology Co., Ltd. (Chengdu, China). UPLC-MS/MS-grade acetonitrile was acquired from Merck Co., Ltd. (Darmstadt, Germany), and UPLC-MS/MS-grade formic acid was obtained from Shanghai Yien Chemical Technology Co., Ltd. (Shanghai, China). LC-MS-grade methanol was acquired from Honeywell International Inc. (New Jersey, NJ, USA). Ultrapure water was acquired from Ultrapure water purchased from Guangzhou Watsons Video Beverage Co. (Guangzhou, China).

### 2.3. Preparation of Stock Solutions of Standards and Samples

#### 2.3.1. Preparation of Standard Stock and Working Solutions

Accurately weighed and dissolved in methanol individually to achieve a concentration of 0.2–0.4 mg/mL, these solutions were then diluted to a concentration of 1 µg/mL. These single standard solutions were used to optimize the MRM parameters. Furthermore, all standards were mixed and diluted to a concentration range of 0.05 µg/mL to 200 µg/mL for constructing calibration curves. All prepared solutions were filtered through a 0.22 µm membrane before being subjected to MS analysis.

#### 2.3.2. Preparation of Samples for UPLC-MS/MS Analysis

All samples were pulverized into powder. Sequentially, 0.1 g of fine powder was accurately weighed and extracted with a 2 mL extracting solution (methanol:water = 7:3) in an ultrasonication water bath at 50 °C for 30 min. After cooling to room temperature, the extraction solution was diluted 10 times with methanol and then filtered through a 0.22 µm PTFE micro-filter before UPLC-MS/MS analysis.

### 2.4. UPLC-MS/MS Conditions

The analytical conditions for the quantitative analysis of 20 ginsenosides were optimized using UPLC-MS/MS (LCMS 8045, Shimadzu, Kyoto, Japan). Instrument acquisition and data processing were performed using LabSolutions LCMS workstation software Ver. 5 (Shimadzu, Kyoto, Japan). The ACQUITY UPLC BEH C18 column (100 mm × 2.1 mm, 1.7 µm) (Waters Co., Ltd., Milford, MA, USA) at 30 °C was used for component separation and quantitative analysis. The mobile phase consisted of 0.1% aqueous formic acid (A) and acetonitrile (B), with a flow rate of 0.3 mL/min. The gradient elution program was as follows: 0.0–1.0 min 28% B, 1.0–11.0 min 28–40% B, 11.0–13.0 min 40–85% B, 13.0–15.0 min 85–95% B, 15.0–17.0 min 95% B, 17.0–17.01 min 95–28% B, and held at 28% B until 23 min. The MS system operated with a positive ESI source in positive mode. The source temperature was maintained at 150 °C, the capillary voltage was set at 3 kV, desolvation gas flow was set at 850 L/h, and desolvation temperature was set at 400 °C. Precursor and product ions were identified for all analytes based on mass-to-abundance ratio. The MRM mode was used for scanning and acquiring mass data for quantification. The MRM parameters for ginsenosides are shown in Appendix A.

### 2.5. Method Validation

Calibration curves for all compounds were constructed by plotting the chromatographic peak area (Y) against the corresponding concentration (X) of the standard solutions. The limits of detection (LOD) and quantification (LOQ) for the indicator components were determined based on mixed reference solution concentrations with signal-to-noise ratios (S/N) of 3 and 10, respectively. Precision was evaluated by continuously injecting the mixed reference solution six times and recording the peak areas. For intra-day and inter-day stability and repeatability tests, a prepared sample solution of 15-year-old forest ginseng leaf was analyzed by measuring the chromatographic peak areas at 0, 2, 4, 6, 12, and 24 h. Additionally, recovery tests were conducted by spiking known amounts of standard solution into the ginsenoside sample solution and comparing the analysis of spiked and unspiked samples. The relative standard deviation (RSD) was used to describe precision, repeatability, and recovery. The results are shown in Appendix A.

### 2.6. Statistical Analysis

The content of each indicator component was calculated in LabSolutions by establishing a standard curve, and the results were expressed as the mean of six replicates ± standard deviation. Using SPSS 26.0 software (SPSS Inc., Chicago, IL, USA), an ANOVA was performed to examine the differences between samples, with a significance level of *p* < 0.05. The data were then imported into Simca-P 14.1 (Umetrics, Umeå, Sweden) for principal component analysis (PCA) and orthogonal partial least squares discriminant analysis (OPLS-DA). Based on the variable importance in projection (VIP) from the OPLS-DA model, biomarkers were screened, with a threshold of VIP > 1 and *p* < 0.05. A cluster heat map was generated according to the biomarkers by TBtools software (version 2.012). Finally, the trends in biomarker changes were analyzed. The content of differential biomarkers before processing was set as 100%, and the percentage change of differential biomarkers after processing was calculated as [percentage change (%) = (average content after processing—average content before processing)/average content before processing × 100%].

### 2.7. Network Pharmacology

In this study, the biomarkers were identified as candidate components. The target proteins of these candidate components were obtained from SwissTargetPrediction (http://www.swisstargetprediction.ch, (accessed on 24 June 2024)). Gene names were then converted using the UniProt database (https://www.uniprot.org, (accessed on 24 June 2024)) to obtain the target proteins of the biomarkers after removing duplicates. Using “against cancer” as the search keywords, target proteins related to anticancer activity were retrieved from the OMIM (https://www.omim.org, (accessed on 24 June 2024)), GeneCards (https://www.genecards.org, (accessed on 24 June 2024)), and TTD (http://db.idrblab.net/ttd, (accessed on 24 June 2024)) databases. The intersection between the target proteins of the biomarkers and the anticancer target proteins was visualized using Venny 2.1.0 (https://bioinfogp.cnb.csic.es/tools/venny, (accessed on 24 June 2024)) to construct a Venn diagram, revealing the common target proteins responsible for the anticancer effects of the biomarkers. The common target proteins were imported into the STRING database (https://string-db.org, (accessed on 24 June 2024)) to construct a protein–protein interaction (PPI) network, which was further analyzed using Cytoscape 3.9.1 software. Important target proteins of biomarkers within the anticancer-related proteins were determined, and the “Network Analyzer” function was utilized to perform topological attribute analysis on the results. Key target proteins were selected based on their degree values exceeding the median. These proteins were subjected to KEGG pathway enrichment analysis using the Metascape database (https://metascape.org, (accessed on 24 June 2024)).

## 3. Results and Discussion

### 3.1. Determination of Ginseng Content before and after Processing

To comprehensively evaluate the content of ginsenosides in *P. ginseng*, we conducted an analysis of the ginsenoside content in different parts with varying cultivation ages and methods before and after processing. The quantitative results of all samples are presented in Appendix A. Initially, a comparison and analysis of the changes in ginsenoside content before and after processing were conducted. Furthermore, the ginsenoside content before and after processing in *P. ginseng* was compared, taking into consideration various cultivation ages and methods. Additionally, a comparison was made of the content and variations of ginsenosides in five parts of three *P. ginseng* varieties after processing. The results of this study provide a deeper understanding of the changes in ginsenoside content in WG and BG and offer guidance for optimizing processing techniques and quality control of *P. ginseng*.

#### 3.1.1. Comparison of Ginsenoside Content before and after Processing

Many studies have found that processing can lead to the production of rare ginsenosides in unprocessed ginseng plants with extremely low or even no ginsenoside content [25]. In cultivated ginseng, the content of Re and Rg1 significantly decreased after processing, while the content of S-Rg3, Rg5, Rk1, Rk3, and F4 significantly increased. In 15-year and 27-year forest ginseng, the content of Re significantly decreased after processing, while the content of S-Rg3, Rg5, R-Rh1, Rk1, Rk3, and F4 significantly increased. Additionally, in cultivated ginseng, the content of prototype ginsenosides Rd and F2 significantly increased in the stem and lateral root after processing. In 15-year forest ginseng, the content of Rd and F2 significantly increased in the leaf, stem, main root, and lateral root after processing. In 27-year forest ginseng, the content of Rd significantly increased in the stem and lateral root after processing, while the content of F2 significantly increased in the main root. According to the Pharmacopoeia of the People’s Republic of China (PPRC), both the leaf and main root parts are used as medicinal parts of ginseng, respectively. The content of rare ginsenosides increased significantly in BG, with the total sum of rare ginsenosides in the leaf of cultivated ginseng, 15-year forest ginseng, and 27-year forest ginseng being 19 times, 18 times, and 33 times higher than in WG, respectively. The total sum of rare ginsenosides in the main root of cultivated ginseng, forest ginseng, and 27-year forest ginseng was 82 times, 56 times, and 106 times higher than in WG, respectively. The results are shown in Appendix A.

Processing has been found to influence the composition of different components in various parts of ginseng. After processing, the levels of Rd increased in both cultivated ginseng and forest ginseng (except for the main root), while the levels of F2 increased in the main root and rhizome. Conversely, the levels of the rare ginsenoside CK decreased in the main root. Comparing 27-year forest ginseng with 15-year forest ginseng revealed that the levels of prototype ginsenosides (Rb1, Rb2, Rb3, Rc, Rd, Rf) in the leaf, stem, rhizome, and lateral root exhibited an increasing trend in a time-dependent manner. However, this increase was more pronounced in the 27-year forest ginseng, suggesting that another type of ginsenoside transforms into prototype ginsenosides during processing. The trends of ginsenoside levels after processing in each part are presented in Appendix A.

#### 3.1.2. Comparison of Ginsenoside Content with Different Cultivation Ages

The relationship between the growth age of forest ginseng and the accumulation of saponins is not proportional [26]. We compared the contents in five parts of 15-year and 27-year forest ginseng, and the results are shown in Figure 4. The results indicated that except for rare ginsenosides in the leaf of BG and prototype ginsenosides in the rhizome of BG, there was no significant difference in the content of forest ginseng between the two age groups. After processing, the content of rare ginsenosides (except S-Rg2) in the leaf of 27-year forest ginseng was significantly higher than that of 15-year ginseng (*p* < 0.05); similarly, the content of prototype ginsenosides (Rb1, Rc, Rd, Re, Rf, Rg1, and F2) in the rhizome of 27-year forest ginseng was significantly higher than that of 15-year ginseng (*p* < 0.05). Conversely, the content of Re in the WG leaf of 15-year forest ginseng was significantly higher than that of 27-year forest ginseng (*p* < 0.01), and other components in different parts of WG were also higher than those of 27-year forest ginseng, such as Rb2, S-Rg2, S-Rg3. However, it is worth noting that although there was no significant difference between the two ages of forest ginseng in WG, the total ginsenoside content of 27-year forest ginseng in BG was higher than that of 15-year forest ginseng, indicating that another type of ginsenoside may be proportional to the age of ginseng.

#### 3.1.3. Comparison of Ginsenoside Content with Different Cultivation Methods

Due to variations in the growth environment, there are differences in the active ingredients between cultivated ginseng and forest ginseng [27]. Research has shown that forest ginseng exhibits a higher abundance and diversity of ocotillol-type ginsenosides, whereas cultivated ginseng has a wider distribution of PPD- and PPT-type ginsenosides [28]. Most of the forest ginsengs sold on the market at present are 15 years old, so the experiments in this section are based on LXS-1 as an example. In this study, significant differences were observed between cultivated ginseng and forest ginseng in terms of the WG leaf and the BG main root and lateral root (Figure 5). The content of Rb1, Rb2, Rb3, Rc, Rd, and Rg1 in the WG leaf of cultivated ginseng was significantly higher than that of forest ginseng (*p* < 0.01). Similarly, the content of Rb1, Rb2, Rb3, Rc, Re, Rg1, Ro, S-Rg3, Rg5, R-Rh1, S-Rh2, Rk1, Rk3, and F4 in the BG main root of cultivated ginseng was significantly higher than that of forest ginseng (*p* < 0.05 or *p* < 0.01). Before processing, the total ginsenoside contents of the leaf and stem parts of YS, respectively, were 115.724 ± 14.844 and 10.3 ± 1.712, whereas those of LXS-1 were 87.738 ± 12.753 and 9 ± 0.602; after processing, the total ginsenoside contents of the leaf and stem parts of YS, respectively, were 117.176 ± 8.782 and 11.28 ± 1.524, while those of LXS-1 were 112.328 ± 17.66 and 9.725 ± 1.151. Both before and after processing, the total ginsenoside content in the aboveground parts (leaf and stem) of cultivated ginseng was higher than that of forest ginseng. However, before processing, the total ginsenoside content in the underground parts (rhizome, main root, and lateral root) of forest ginseng was 1.2–1.8 times that of cultivated ginseng, whereas after processing, the total ginsenoside content in the main root and lateral root of forest ginseng was lower than that of cultivated ginseng (except for the rhizome). Importantly, after processing, the main root of cultivated ginseng exhibited a significantly higher ginsenoside content compared to forest ginseng.

#### 3.1.4. Comparison of Ginsenoside Content in Different Parts of Black Ginseng

The different parts of TCM (traditional Chinese medicine) materials often exhibit variations in chemical composition and content, which may lead to differences in clinical efficacy and applications [29,30]. The 2020 edition of the PPRC considers the main root and leaf of *P. ginseng* as distinct medicinal materials. We compared the contents of five parts of ginseng and generated a cluster heatmap. The results, shown in Appendix A, indicate that the five parts can be well distinguished and clustered into two major groups: the aboveground parts and the underground parts. The cluster heatmap revealed that the content of the determination index (Rb1, Re, Rg1) in the PPRC was higher in *P. ginseng*. The aboveground parts of WG were primarily enriched with Re and Rg1, with F2 and F3 being notably high only in the leaf part. Conversely, the underground parts of WG were rich in Rb1, with a content ranking showing a trend of Rb1 > Re > Rg1 [31]. In BG, the levels of Re, F4, and Rk1 were higher in the aboveground parts, whereas Rb1, Rk1, and Rg5 were more abundant in the underground parts. These findings provide valuable insights for the extraction and selection of specific ginsenoside components from different parts.

To further determine the changes and differences in ginsenoside content among different parts of BG, one-way ANOVA analysis was performed on the components that significantly increased or decreased after processing. The results used a–e to indicate significant differences at *p* < 0.05, with the same letter indicating no significant difference (Figure 6). The overall trend showed that Rb1 had the highest content among the prototype ginsenosides in BG [32], while Rk1 had the highest content among the rare ginsenosides [33], and both were most present in the lateral root. The content of Rb1 exhibited large disparities across the five parts, with the lateral root exhibiting the highest concentration. Furthermore, significant differences were observed between the lateral root, rhizome, and the remaining three parts. The content of Re was highest in the leaf, with significant differences between the leaf and other parts. The order of Rk1 content among different parts was lateral root > rhizome ≈ main root > leaf > stem, with significant differences among all parts. Significant differences in rare ginsenosides were found in both the leaves and main roots (except for S-Rg3), indicating that both the leaf and main root can serve as sources for these two types of medicinal materials after processing.

From the above results, it is observed that the contents and types of ginsenoside in black ginseng considerably increase. Specifically, there is an increase in the content of rare ginsenosides in black ginseng, including Rg5, Rk1, and Rk3, which were either not detected (below the LOD) or detected in trace amounts (below the LOQ) in the white ginseng. Conversely, the content of prototype ginsenosides decreases. According to research [17], the prototype ginsenosides changed the dammarane skeleton through demalonylation, decarboxylation, deglycosylation, and dehydration reactions when under high temperature and high pressure. Consequently, these reactions result in the conversion of the prototype ginsenosides into rare ginsenosides. This explains the increase in the content of trace rare ginsenosides after processing.

Theoretically, it can be posited that the content of the prototype ginsenosides is expected to decline after the processing, ultimately resulting in their transformation into the corresponding rare ginsenosides [25,34]. However, in this study, prototype ginsenosides such as Rb1, Rb2, Rb3, Rd, and Rf, exhibited contrasting trends in different parts. It was hypothesized that the cause behind the observed changes in the trends of Rd, F2, and CK may be attributed to the thickness of the main roots, impeding the infiltration of thermal energy during the process of steaming, consequently leading to a lower degree of steaming in comparison to the leaf and stem. Combined with the deglycosylation pattern, it is postulated that Rd in the main root first loses one sugar group at C-3 to form F2 during steaming but has difficulty losing another sugar group to form CK, and it is possible for CK to transform other substances. Thus, there was a decrease in the content of Rd, an increase in the content of F2, and a decrease in the content of CK. The changes in ginsenosides after processing in other parts are like those reported in the study by Chen et al. [35].

Ginsenosides can be classified into neutral ginsenosides and acidic ginsenosides. Malonyl ginsenosides, a type of acidic ginsenoside, are highly polar, water-soluble compounds with strong hydrophilic properties [36]. They are the predominant form of ginsenosides found in fresh ginseng and white ginseng [37]. Malonyl ginsenosides have been shown to possess therapeutic effects in the treatment of diabetes and neuroprotection [38]. Previous studies have suggested that high temperature is an essential factor influencing the degradation of malonyl ginsenosides occurring moieties [37]. It has been found that even high temperature cannot degrade malonyl ginsenoside under dry condition. The main reason is that under high temperature, malonyl ginsenoside dissolved in water will incur decarboxylation, depropionylation, and deacetylation reactions, generating malonyl acid, acetic acid, and the corresponding neutral saponins, and then under the hydrolysis of malonyl acid and acetic acid, the neutral ginsenoside is further converted into rare ginsenosides [39]. In this study, it was noted that the contents of prototype ginsenosides exhibited an increase after processing, which can plausibly be ascribed to the hydrolysis of malonyl ginsenosides under high-temperature conditions. Because of that, this transformation resulted in the removal of the malonyl group and the concomitant formation of neutral ginsenosides (i.e., some prototype ginsenosides).

In addition, the content of malonyl ginsenosides varies in *P. ginseng* of different growth ages and cultivation methods [27]. Shin et al. [40] found that compared to the 6-year ginseng root, the content of malonyl ginsenosides was higher than that in 5-year ginseng root. However, quantitative analyses correlating the growth age of forest-grown ginseng with malonyl ginsenoside content are currently limited. Equally, this pattern was only noted in Rb1, Rb2, Rb3, Rc, Rd, Rf, and F2, whereas the remaining four prototype ginsenosides did not manifest such conduct. Given the absence of quantitative analysis of malonyl ginsenosides in this experiment, a definitive conclusion cannot be reached with utmost precision. Further investigations are intended to be carried out to improve the analysis of malonyl ginsenoside content in various samples.

Previous research reports have indicated that there is a turning point in the growth and accumulation of bioactive components in ginseng at around 15 years, with most ginsenosides accumulating between 5 and 15 years and gradually decreasing after 15 years [41]. This may be related to the low expression of terpene skeleton biosynthesis genes in ginseng over 15 years old, leading to a decrease in ginsenoside content. Fang et al. [21] conducted high-throughput transcriptome sequencing on forest ginseng and cultivated ginseng at different ages and found that differentially expressed genes (DEGs) related to ginsenoside synthesis had a relatively small impact on forest ginseng. Among the enzymes involved in ginsenoside synthesis, only hydroxymethylglutaryl-CoA synthase, 1-deoxy-D-xylulose-5-phosphate synthase, squalene epoxidase, cytochrome P450, and UDP-glycosyltransferases were highly expressed in the roots of forest ginseng, indicating that the growth age had little effect on ginsenoside synthesis in forest ginseng. In addition, Zhang et al. [42] found that the accumulation of ginsenoside content in ginseng leaves did not necessarily follow a consistent trend with increasing growth age. In this study, there was a small difference in ginsenoside content between the two age groups of forest ginseng before and after processing, suggesting that they can be used interchangeably in practical applications.

Ginseng cultivated in gardens can typically be harvested after approximately 5 years [43], while the harvesting age of forest ginseng varies greatly, ranging from decades to hundreds of years. Due to its slower growth and longer lifespan, forest ginseng is believed to possess superior medicinal qualities compared to cultivated ginseng [28]. In this study, it was found that the total ginsenoside content in the root of forest ginseng in WG was 1.2–1.8 times that of cultivated ginseng root. Ma et al. [44] conducted a comprehensive and systematic comparison of secondary metabolite biosynthesis between forest and cultivated ginseng and found that forest ginseng accumulates more ginsenosides. This is related to the precursor substances involved in ginsenoside synthesis in forest ginseng, which experiences greater growth stress compared to cultivated ginseng. As a result, more enzymes involved in ginsenoside synthesis are induced in forest ginseng, leading to higher levels of ginsenoside precursors such as 2,3-oxidation-derivative of dammarenediol-II. However, in this study, the ginsenoside content in the cultivated ginseng leaves of WG was significantly higher than that in the forest ginseng leaves of WG. This may be related to the DEGs in the leaves of cultivated ginseng and forest ginseng. Fang et al. [21] found that cultivated ginseng had the highest number of DEGs in the leaf, while forest ginseng had the highest number of DEGs in the root and the lowest number in the leaf, indicating variations in ginsenoside synthesis processes among different varieties and plant parts. Additionally, these differences may also be influenced by soil microorganisms and light conditions in the growth environment [45,46,47].

Significant differences in the content of ginsenosides among different parts have also been observed [8]. In this experiment, the components with higher content in different parts in WG and BG were also found to be different. According to the Chinese Pharmacopoeia, Rb1, Rg1, and Re are designated as the index components for *P. ginseng* root, whereas Rb1 is excluded from the index components for *P. ginseng* leaf, which aligns with the findings of this research. Therefore, based on the comprehensive findings of this study, it can be considered feasible to use F4 as the index component for black ginseng leaf and Rb1 and Rk1 as the index components for black ginseng root.

### 3.2. Exploration of Biomarkers of Ginseng before and after Processing Based on Chemometric Analysis

The data of ginseng samples were analyzed using PCA. From the PCA score plot (Appendix A), it can be observed that the sample clusters in WG and BG were separated into two groups, suggesting that the established method effectively characterizes the differences. Furthermore, the OPLS-DA analysis was conducted on cultivated ginseng and forest ginseng samples, respectively, with the aim of identifying the different biomarkers that can be used to distinguish WG and BG. The results are shown in Figure 7. The R2X (cum) values of the OPLS-DA models for the three varieties range from 0.988 to 0.997, the R2Y (cum) values range from 0.813 to 0.879, and the Q2 (cum) values range from 0.835 to 0.88, all of which are greater than 0.5. This indicates that the established models have a good discriminative ability for inter-group samples and high aggregation for samples within the same group. The permutation test revealed no evidence of overfitting, as supported by the intercept values for R2 and Q2 being less than 0.4 and less than 0.05, respectively.

The designation of VIP > 1 and *p* < 0.05 as biomarkers is widely acknowledged in the field. The biomarkers for the three varieties and their ranking are as follows: biomarkers of cultivated ginseng after processing were screened out, with the significance ranking as R-Rh1 > Rk3 > F4 > S-Rg3 > Rg5 > Rk1; biomarkers of 15-year forest ginseng after processing were screened out, with the significance ranking as R-Rh1 > Rk3 > F4 > S-Rg3 > Rg5 > Rk1; biomarkers of 27-year forest ginseng after processing were screened out, with the significance ranking as R-Rh1 > Rk3 > Rg5 > S-Rg3 > Rk1 > F4. These biomarkers are speculated to be able to distinguish the differences between white ginseng and black ginseng.

To systematically evaluate biomarkers, a cluster heatmap was generated, as shown in Appendix A. The samples from the three varieties were classified into two distinct categories: WG and BG. Additionally, the five parts were categorized into two groups: the aboveground parts forming one class and the underground parts forming another. Notably, some WG underground parts from 27-year forest ginseng clustered with BG.

Before processing, the content of R-Rh1 was higher in all sites, and the content of S-Rg3 was higher in the leaf and lateral root. After processing, the aboveground parts were higher in F4, while the underground parts were higher in three biomarkers (Rk1, S-Rg3. and Rg5). Hence, these six compounds can effectively differentiate between ginseng before and after processing and are considered differential markers.

### 3.3. Percentage Changes of Biomarkers of Ginseng before and after Processing

An analysis of the percentage change of Rg3, Rg5, Rk1, F4, Rk3, and R-Rh1 across different samples showed that their increase in the leaf and stem of both cultivated and forest ginseng was more significant than in the main root. Notably, Rk1’s and Rk3’s percentage increases in the leaf of forest ginseng were substantially higher compared to the main root, with other components also showing increases. In addition, the percentage changes of the index components Rb1, Re, and Rg1 of ginseng in the pharmacopoeia were analyzed, and the results showed that the percentage changes of these three components in the main roots of cultivated ginseng and wild ginseng were relatively large, ranging from −91.5% to −99.9% (Appendix A). The percentage changes of these rare ginsenosides in various parts are detailed in Figure 8A–C. When comparing cultivation methods, the increase in rare ginsenosides was generally higher in cultivated ginseng, except for the leaf and stem, as opposed to forest ginseng (LXS-1), shown in Figure 8D. Between 15-year and 27-year forest ginseng, the younger ginseng showed a higher percentage change in most parts, excluding the main root, as indicated in Figure 8E.

The percentage changes of rare ginsenosides in *P. ginseng* leaves or stems were greater than in the main root, likely because the leaves and stems, being thinner and more delicate, allow heat to penetrate more easily during steaming. This results in a more complete conversion of prototype ginsenosides through deglycosylation and dehydration reactions. It’s crucial to consider biomass variability when exploring processing methods. Additionally, the ginseng leaf contains a higher ginsenoside content than the main root [42], providing a larger source for generating rare ginsenosides. Thus, after processing, the leaf’s rare ginsenoside content is relatively higher in the leaf compared to the main root.

Environmental factors and ginseng development are closely linked [45]. The percentage change of rare ginsenosides in cultivated ginseng parts, excluding the leaves, was higher than in forest ginseng, possibly due to different growth environments promoting more malonyl ginsenoside accumulation in forest ginseng leaves [3]. However, the limited sample size and potential environmental influences warrant further research.

The leaf of 15-year forest ginseng showed a significantly higher quantity of ginsenosides after processing compared to the 27-year forest ginseng. This is due to a greater conversion to rare ginsenosides upon processing. thus resulting in a greater percentage change of rare ginsenosides in 15-year forest ginseng compared to 27-year forest ginseng. However, the total ginsenoside content in other parts of 27-year forest ginseng (except for the roots) was slightly higher than that in 15-year forest ginseng in WG. On the other hand, its percentage change in ginsenoside content was not as significant as that observed in 15-year forest ginseng. Factors such as biomass, the content of malonyl ginsenoside, the environment, and the age of the ginseng plant may all play a role in these variations. To summarize, to increase the production of rare ginsenosides, it is recommended to choose the economically feasible stem and leaf parts of cultivated ginseng, as these parts exhibit a higher percentage change of rare ginsenosides compared to the more costly root parts.

### 3.4. Prediction of Anti-Cancer Mechanisms of Ginseng Biomarkers Based on Network Pharmacology

Rg3, Rg5, Rk1, F4, Rk3, and R-Rh1 were selected as candidate components of black ginseng for network pharmacology analysis to investigate their anti-cancer mechanisms.

#### 3.4.1. Target Screening and PPI Network Construction

By searching SwissTargetPrediction, a total of 69 target proteins were obtained from biomarkers. Additionally, 2713 target proteins related to anticancer activities were identified through screening the OMIM, Gene Cards, and TTD databases. The intersection of the biomarkers-related target proteins and anticancer-related target proteins resulted in a total of 61 common target proteins of black ginseng. The common target proteins were input into the String database for PPI network analysis. Using Cytoscape software (Ver. 3.10.0), a total of 28 key target proteins of black ginseng were selected, and the protein interactions were analyzed (Appendix A).

#### 3.4.2. KEGG Pathway Enrichment Analysis

In the KEGG enrichment analysis, a total of 122 main pathways of biomarkers were collected. Some of the pathways that were predominantly involved include pathways in cancer, the MAPK signaling pathway, EGFR tyrosine kinase inhibitor resistance, the PI3K-Akt signaling pathway, the TNF signaling pathway, etc. The KEGG database was used to interpret the pathways in cancer pathway (Figure 9). The results showed that the pink boxes contained proteins with anticancer effects in black ginseng. Based on the results, we inferred that the EGFR/PI3K/Akt/mTOR pathway, with TGFα as the initial target, may be a crucial pathway for the anticancer effects of black ginseng. Experimental evidence has demonstrated that Rg5 can inhibit the Akt signaling pathway, downregulate the expression of Bcl-2, and thereby promote apoptosis and autophagy in retinoblastoma cells and breast cancer cells [48,49]. Ginsenoside Rg3 exhibited anticancer activity in lung cancer cells, hepatocellular carcinoma cells, and gastric cancer cells by inhibiting the PI3K/Akt/mTOR signaling pathway [50].

However, it should be noted that network pharmacology can only provide preliminary predictions regarding the relevant target proteins and pathways of biomarkers. Further validation is required at the cellular and animal levels.

## 4. Conclusions

A simple and feasible UPLC-MS/MS method for the simultaneous determination of 20 ginsenosides was established. The contents and species of rare ginsenosides in various parts of cultivated ginseng and forest ginseng increased after processing, which provided guidance for the extraction and screening of specific ginsenoside components from different parts. Based on the differences in ginsenoside contents before and after processing, the corresponding differential biomarkers (Rg3, Rg5, Rk1, F4, Rk3, and R-Rh1) were screened out, which can be used as the quality control indexes of black ginseng and the indexes of containing measurements, and meanwhile, it was revealed that there was a significant difference in the conversion rate of the processing biomarkers in different parts of ginseng, which provides a reference for the comprehensive utilization of ginseng. Finally, the network pharmacological analysis of the processing biomarkers predicted that the key anti-tumor pathway of black ginseng was EGFR/PI3K/Akt/mTOR.

## Figures and Tables

**Figure 1 foods-13-02497-f001:**
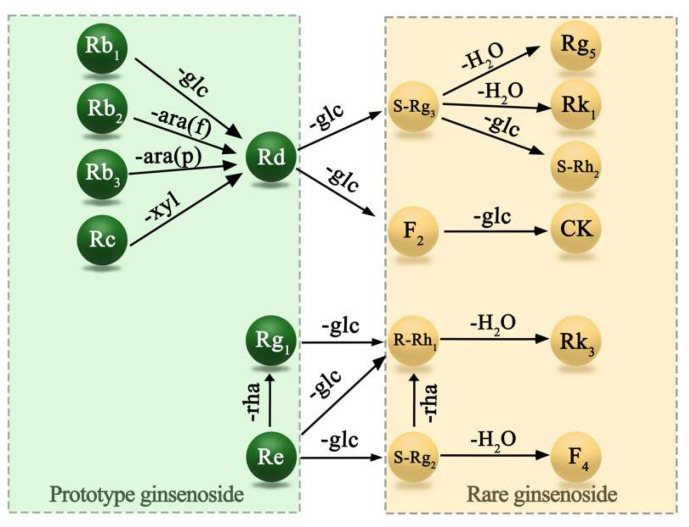
The transformation of ginsenosides. ara(f), α-L-arabinofuranosyl; ara(p), α-L-arabinopyranosyl; glc, β-D-glucopyranosyl; xyl, β-L-xylopyranosyl; rha, a-L-ahamnopyranosyl.

**Figure 2 foods-13-02497-f002:**
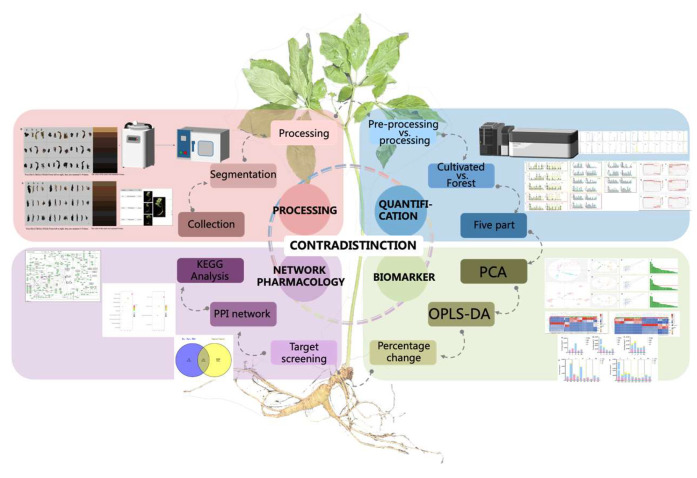
The overall workflow of this research.

**Figure 3 foods-13-02497-f003:**
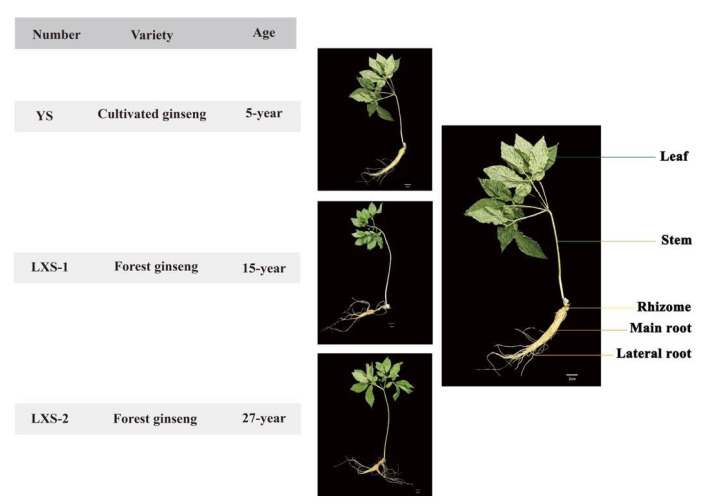
*P. ginseng* sample information.

**Figure 4 foods-13-02497-f004:**
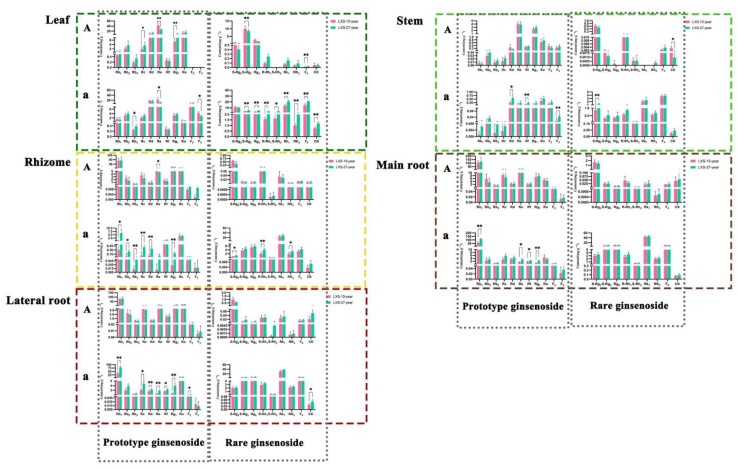
Comparison of the content of five parts of forest ginseng in 15 years and 27 years. The capital letters represent before processing, and lowercase letters represent after processing. Compared to the LXS-27: * *p* < 0.05, ** *p* < 0.01

**Figure 5 foods-13-02497-f005:**
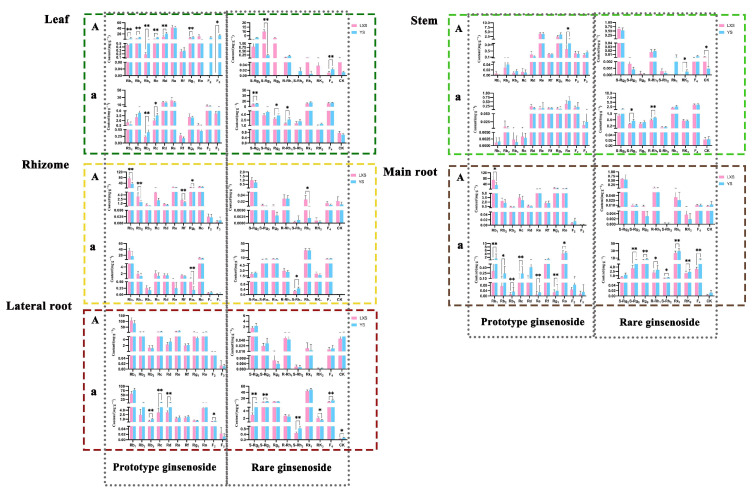
The contents of five parts of cultivated ginseng and forest ginseng. The capital letters represent before processing, and lowercase letters represent after processing. Compared to the YS: * *p* < 0.05, ** *p* < 0.01.

**Figure 6 foods-13-02497-f006:**
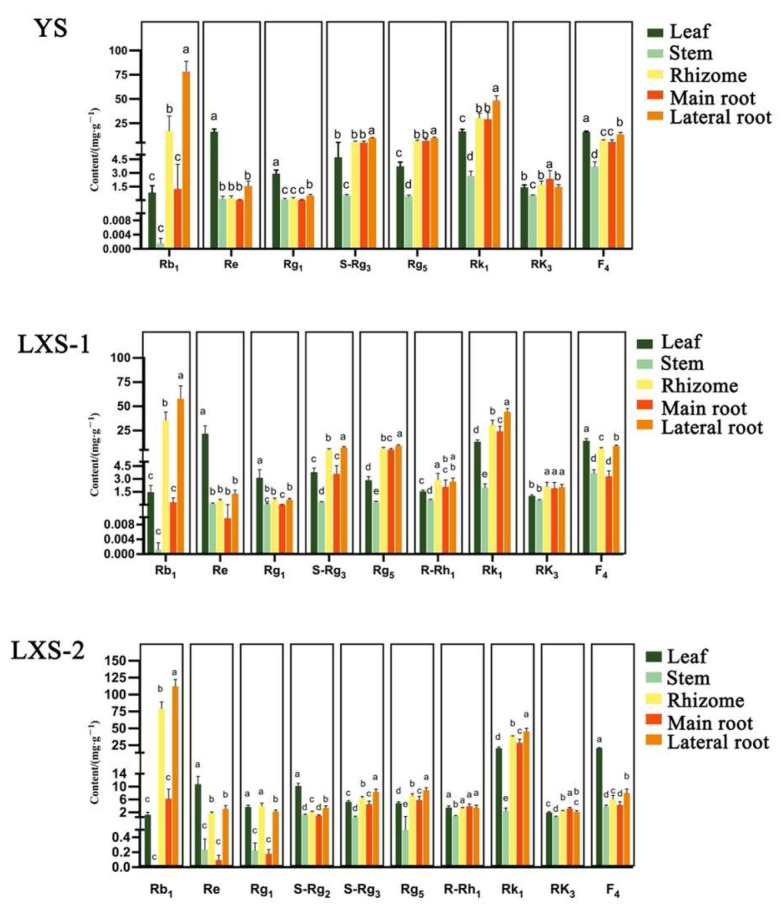
Comparison of ginsenoside content in different parts of black ginseng. The results used a–e to indicate significant differences at *p* < 0.05, with the same letter indicating no significant difference.

**Figure 7 foods-13-02497-f007:**
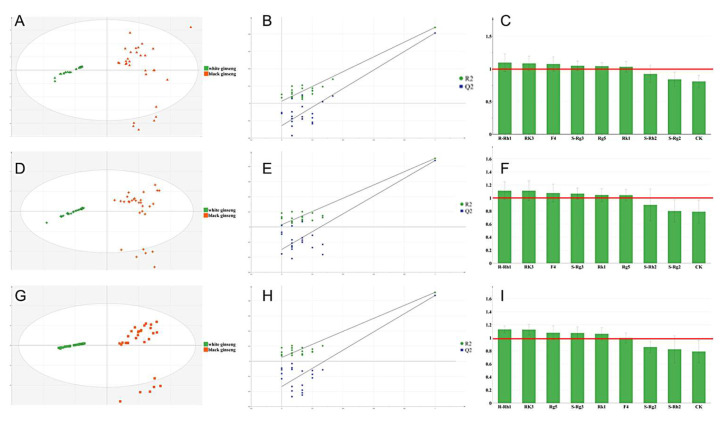
Score plots (**A**,**D**,**G**), replacement plots (**B**,**E**,**H**)**,** and VIP value plots (**C**,**F**,**I**) for OPLS-DA analysis of YS, LXS-1, and LXS-2, respectively.

**Figure 8 foods-13-02497-f008:**
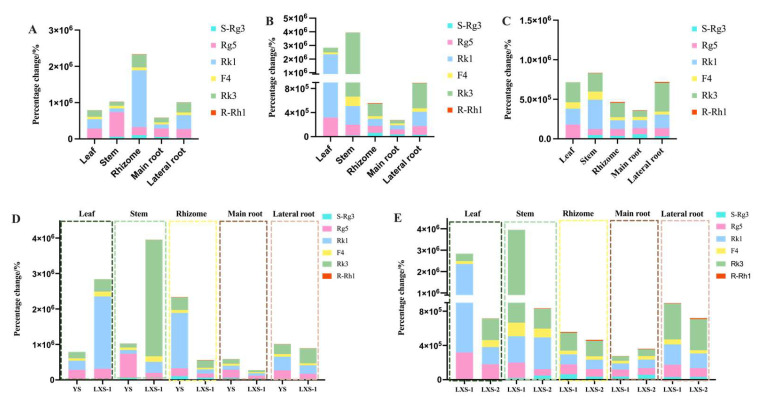
Percentage change of biomarkers in different parts of YS (**A**), LXS-1 (**B**), and LXS-2 (**C**) after processing; percentage change of biomarkers in different parts of YS and LXS-1 (**D**); percentage change of biomarkers in different parts of LXS of different age after processing (**E**).

**Figure 9 foods-13-02497-f009:**
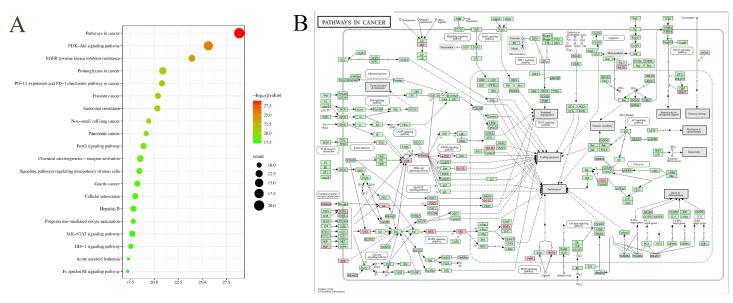
KEGG enrichment (**A**) and targets of biomarkers and the pathways in cancer pathways (**B**).

## Data Availability

The original contributions presented in the study are included in the article/Appendix A, further inquiries can be directed to the corresponding author.

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
