# Peer review of "Integrated Analysis of Ginsenoside Content and Biomarker Changes in Processed Ginseng: Implications for Anti-Cancer Mechanisms"

_foods, 2024, doi:10.3390/foods13162497_

Round 1
Reviewer 1 Report
Comments and Suggestions for Authors
In this study, a large experimental data set was derived by analyzing the contents of 20 ginseng saponins in the five parts (leaf, stem, rhizome, main root, lateral root) of three samples (YS, LXS-1, -2), and two types of processing (WG, BG). In addition, genes and markers for representative six markers of BG that can exhibit anti-cancer activity were extracted, and the pathways for anti-cancer activity were presented.
Since there are many previous research results on ginseng ingredients, the results of this study can be predicted to some extent; the saponin contents vary depending on the habitat/ages and the parts. In particular, when BG is manufactured by heat treatment from fresh ginseng or WG, major (prototype) saponins are well-known to be converted to minor (rare) saponins and the rare saponins are increased mainly by hydrolysis (deglycosylation) from major prototypes.
Therefore, the value of this study seems to be meaningful and valuable in presenting actual changes in the content of 20 saponins from various samples/parts and the characteristic markers of BG (F4:leaf, RB1 & Rk1:root) from the data sets & results.
However, explanation/corrections are needed to the questions about of the results as shown below for acceptance in the journal, and correction of typographical errors are required.
-In 3.1.2 – 3.1.3, the total ginseng saponin content of each of the three samples should be added to the table for easy understanding.
-In lines of 262-279, which forest ginseng used in lines 3.1.3 & fig 8? 15 years or 27 years or average of two?
-In lines 269-270, Rb1 was also significantly higher in WG than in forest ginsengs (refer suppl table s4)
-In lines 270-271, it was mentioned that the contents of prototype saponin in cultivated ginseng was significantly higher than in forest ginsengs. When the contents of total saponins depends on the amount of high-content main saponins, it does not consistent that the content of total saponin in the underground of LXS was twice that of cultivated ginseng in lines 274-278.
-In lines 270-272, the part was called root, but it’s not clear which part of the underground part it was, or the sum of the three parts (rhizome, main, lateral root).
-In case of Rb1, the cultivated one was not higher than the forest one in the results for each underground part (refer suppl table s4).
-In lines 272-274, the total saponin content of each type should be clearly put the numbers in the text and suppl tables.
-In addition, in lines 274-278, the total saponin content of the underground part of LXS was about twice as high as that of cultivated ginseng, but the contents were lower than that of YS after black ginseng process. As the conversion of prototype ginsenosides to rare ginsenosides progressed, sugars were lost and the molecular weight becomes smaller, resulting in a lower content. Therefore, it’s necessary to mention of this point in the text and check if there are any references that mention this point.
-In lines 340-361, malonyl ginsenosides as mentioned in lines 342-343, are found to be main ones in fresh ginseng when directly extracted, however, these saponins are easily hydrolyzed even at about 50 degrees during the extraction/concentration process. So, in lines 352-361, malonyl ginsenosides may not be detected because these were easily hydrolyzed in WG process (60 degrees). please find the reference of this point and mention it in the text.
-In lines 382-383, total saponins of the roots in the WG of forest ginseng were 1.2 times higher than that of YS. However, the total saponins of the underground part of forest ginseng before processing was about twice of YS in lines 274-278 ... Why are these two contents expressed differently?
In Fig. 14B, from the extraction of anticancer activity pathways targeting key markers of BG, 28 key target proteins were extracted and the EGFR/PI3K/Akt/mTOR pathway was presented, it would be better to present these pathways in a simplified manner with more emphasis. It’s difficult to understand because the map was too complex and low resolution.
-In 3.3, is LXS the average of LXS-1 and – 2?
-In fig13-4, the saponin content of leaves and underground parts was large on average, so changes in their composition and content are seen as important, but it can be misunderstood that the large content change in the stem or rhizome, which has a very low content, seemed significant. It needs to mention that the change of major saponins was important in the text.
In addition, please correct typographical errors as belows.
In line 26, Panax ginseng should be italic
in lines of 192, 197, 199, 287, 291, 352, 402 P. ginseng should be italic
in line 96 60℃9 -> 60℃
spacing errors
in line 123, 2ml -> 2 ml
in line 131, 100mm x 2.1mm, 1.7um -> 100 mm x 2.1 mm, 1.7 um
Author Response
Comments 1: In 3.1.2 – 3.1.3, the total ginseng saponin content of each of the three samples should be added to the table for easy understanding.
Response 1: Thank you for pointing this out. We agree with this comment. Therefore, we have added the total ginsenoside content of the three samples to the Supplementary Material Table S4.
Comments 2: In lines of 262-279, which forest ginseng used in lines 3.1.3 & fig 8? 15 years or 27 years or average of two?
Response 2: Thank you for pointing this out. We have, accordingly, revised this expression, in lines 268-269 of the manuscript.
Comments 3: In lines 269-270, Rb1 was also significantly higher in WG than in forest ginsengs (refer suppl table s4)
Response 3: Thank you for pointing this out. After revisiting the data, which was an oversight by us, we have corrected this result and have made changes on line 271 and in Figure 8.
Comments 4: In lines 270-271, it was mentioned that the contents of prototype saponin in cultivated ginseng was significantly higher than in forest ginsengs. When the contents of total saponins depends on the amount of high-content main saponins, it does not consistent that the content of total saponin in the underground of LXS was twice that of cultivated ginseng in lines 274-278.
Response 4: The reference in lines 270-271 refers to a wider distribution of ginsenosides contained in cultivated ginseng, which showed that cultivated ginseng contained a greater variety of ginsenosides rather than a higher content. The reference also mentioned that the content of OC-type ginsenosides was higher in forest ginseng, so it was considered to be consistent with the results of this study.
Comments 5: In lines 270-272, the part was called root, but it’s not clear which part of the underground part it was, or the sum of the three parts (rhizome, main, lateral root).
Response 5: Thank you for pointing this out. Therefore, we modify the expression to emphasize this point. Also add the results of this section, as described in lines 273-275.
Comments 6: In case of Rb1, the cultivated one was not higher than the forest one in the results for each underground part (refer suppl table s4).
Response 6: In 3.13, this part of the study was a comparison of ginsenoside content between cultivated ginseng (YS) and 15-year forest ginseng (LXS-1), and according to Table S4 in the Supplementary Material, Rb1 in the cultivated ginseng in the BG had a higher content of Rb1 in the cultivated ginseng than in the forest ginseng in the BG. The same was observed for the other ginsenosides.
Comments 7: In lines 272-274, the total saponin content of each type should be clearly put the numbers in the text and suppl tables.
Response 7: Thank you for pointing this out. We have therefore added this data to lines 272-279 of the manuscript and to Table S4.
Comments 8: In addition, in lines 274-278, the total saponin content of the underground part of LXS was about twice as high as that of cultivated ginseng, but the contents were lower than that of YS after black ginseng process. As the conversion of prototype ginsenosides to rare ginsenosides progressed, sugars were lost and the molecular weight becomes smaller, resulting in a lower content. Therefore, it’s necessary to mention of this point in the text and check if there are any references that mention this point.
Response 8: Thank you for your careful guidance, which is explained in lines 49-57 and 336-341 in the manuscript, and in reference 17.
Comments 9: In lines 340-361, malonyl ginsenosides as mentioned in lines 342-343, are found to be main ones in fresh ginseng when directly extracted, however, these saponins are easily hydrolyzed even at about 50 degrees during the extraction/concentration process. So, in lines 352-361, malonyl ginsenosides may not be detected because these were easily hydrolyzed in WG process (60 degrees). please find the reference of this point and mention it in the text.
Response 9:
1.Malonyl ginsenosides are indeed hydrolyzed during the preparation of white ginseng, but due to the lower temperature hydrolysis is not high and hydrolysis is not clear, and with the increase in temperature and length of concoction, it is possible that they will continue to be hydrolyzed to produce the prototypical ginsenosides[with references: Fan, J.; et al. Comprehensive Investigation of Ginsenosides in the Steamed Panax quinquefolius with Different Processing Conditions Using LC-MS. Molecules 2024.].
2. It has been found that even high temperature cannot degrade malonyl ginsenoside under dry condition, the main reason is that under high temperature, malonyl ginsenoside dissolved in water will occur decarboxylation, depropionylation and deacetylation reaction, generating malonyl acid, acetic acid and the corresponding neutral saponins, and then under the hydrolysis of malonyl acid and acetic acid, the neutral ginsenoside is further converted into rare ginsenosides.[with references: Liu, Z.; et al. Remarkable Impact of Acidic Ginsenosides and Organic Acids on Ginsenoside Transformation from Fresh Ginseng to Red Ginseng. J Agric Food Chem. 2016] Therefore, with the gradual decrease of water in the drying process of white ginseng, the malonyl ginsenosides will be gradually stabilized and no longer degraded, and will be degraded again when encountered with water vapor in the steaming process of black ginseng.
The above is referred to in lines 365-371 of the manuscript.
Comments 10: In lines 382-383, total saponins of the roots in the WG of forest ginseng were 1.2 times higher than that of YS. However, the total saponins of the underground part of forest ginseng before processing was about twice of YS in lines 274-278 ... Why are these two contents expressed differently?
Response 10: Thank you for pointing this out. We have reworked the expression, see lines 286 and 407-408 of the manuscript.
Comments 11: In Fig. 14B, from the extraction of anticancer activity pathways targeting key markers of BG, 28 key target proteins were extracted and the EGFR/PI3K/Akt/mTOR pathway was presented, it would be better to present these pathways in a simplified manner with more emphasis. It’s difficult to understand because the map was too complex and low resolution.
Response 11: Thanks for the advice. Since the figure is the pathway involved in the relevant target of black ginseng anti-tumor effect, the pink box is the protein acting on the pathway, which is connected with both upstream and downstream, so the figure looks more complicated, we will improve the resolution of the picture and update it as much as possible.
Comments 12: In 3.3, is LXS the average of LXS-1 and-2?
Response 12: Most of the forest ginsengs sold on the market at present are 15 years old, so the experiments in this section are based on LXS-1 as an example. We have, accordingly, revised this expression, in lines 480 and Figure 11-D of the manuscript.
Comments 13: In fig13-4, the saponin content of leaves and underground parts was large on average, so changes in their composition and content are seen as important, but it can be misunderstood that the large content change in the stem or rhizome, which has a very low content, seemed significant. It needs to mention that the change of major saponins was important in the text.
Response 13: Thanks for the advice. We have mentioned the percentage change of the main ginsenosides in the text (line 473-476) and put the relevant data in the Supplementary Material (Table S6).
Comments 14: In addition, please correct typographical errors and spacing errors as belows.
In line 26, Panax ginseng should be italic; in lines of 192, 197, 199, 287, 291, 352, 402 P. ginseng should be italic; in line 96 60℃9 -> 60℃; in line 123, 2ml -> 2 ml; in line 131, 100mm x 2.1mm, 1.7um -> 100 mm x 2.1 mm, 1.7 um.
Response 14:Thank you for pointing this out. We have corrected typographical errors and spacing errors. This change can be found line94, 96, 127, 135, 136, 196, 201, 203, 205, 301, 305, 376, 426, 427.
Special thanks to you for your good comments.

Reviewer 2 Report
Comments and Suggestions for Authors
In this study, Authors investigated the effects of processing on ginsenosides in different parts of cultivated and forest ginseng. Moreover, a sum of eleven prototype ginsenosides and nine rare ginsenosides were screened as the measured indicator components. Therefore I highly appreciate work done but manuscript’s Authors should make the essential additions and corrections.
- Please, use a distance between the word and the bracket, e.g. line 30 “ginseng[2]” - it should be: “ginseng [2]”.
- Line 94, 192 and another - Please use italics for Latin names (P.ginseng).
- Line 96 - Please correct the typo.
- line 96-97 - Why did the Authors use different drying temperatures for white and black ginseng group?
- Subsection 2.2 - There is no information about methanol (preparation of standard solution - line 115).
- Line 123 - What was the purity of water using as extraction solution?
- Line 124 - Is the ultrasonication water bath for 30 min doesn’t change the temperature of extraction process?
Figure 4, 6, 7, 8, 9, 12 and 14 - they are unreadable in current form. I recommend moving some of them to the supplementary material.
Best regards
Author Response
Comments 1: Please, use a distance between the word and the bracket, e.g. line 30 “ginseng[2]” - it should be: “ginseng [2]”.
Response 1: Thank you for pointing this out. We have corrected such errors and updated them in the manuscript.
Comments 2: Line 94, 192 and another - Please use italics for Latin names (P.ginseng).
Response 2: Thank you for pointing this out. We have corrected such errors and updated them in the manuscript. This change can be found line94, 196, 201, 203, 205, 301, 305, 376, 426, 427.
Comments 3: Line 96 - Please correct the typo.
Response 3: Thank you for pointing this out. We have corrected this errors and updated them in the manuscript.
Comments 4: Why did the Authors use different drying temperatures for white and black ginseng group?
Response 4: Thank you for pointing this out. It's a typographical error. We have corrected it.
Comments 5: Subsection 2.2 - There is no information about methanol (preparation of standard solution - line 115).
Response 5: Thank you for pointing this out. We have added complete information in section 2.2.
Comments 6: Line 123 - What was the purity of water using as extraction solution?
Response 6: Thank you for pointing this out. We have added complete information in section 2.2.
Comments 7: Line 124 - Is the ultrasonication water bath for 30 min doesn’t change the temperature of extraction process?
Response 7: The temperature does not change because the ultrasonic cleaner used can be set to a constant temperature and the sample can be extracted at a constant temperature.
Comments 8: Figure 4, 6, 7, 8, 9, 12 and 14 - they are unreadable in current form. I recommend moving some of them to the supplementary material.
Response 8: Thanks to your suggestion, we have moved figures 9 and 12 to the supplementary material.
Special thanks to you for your good comments.

Round 2
Reviewer 1 Report
Comments and Suggestions for Authors
Authors revised the manuscript faithfully to reflect reviewer's comments.
It canbe acceptable for publication.
Author Response
Dear Editor and Reviewer,
Thank you very much for taking the time to review this manuscript again. We thank the reviewers for the time and effort that they have put into reviewing the previous version of the manuscript. The suggestions have enabled us to improve our work.
Thanks again!
Sincerely,
Xin He

Reviewer 2 Report
Comments and Suggestions for Authors
Dear Authors,
Thank you so much for your corrections. At the same time, I would like to ask you to correct Figure 4, 5 and 6, because they are still unreadable.
Best regards,
Reviewer
Author Response
Dear Editor and Reviewer:
Thank you very much for taking the time to review this manuscript again. And thank you for your constructive comments on my manuscript. We have put Figures 4-6 into the Supplementary Materials.Based on the revisions, we have uploaded a revised version of the document.
Thanks again for all your hard work!We hope that the revised manuscript is accepted for publication in the Foods.
Sincerely,
Xin He
